# Parietal alpha tACS shows inconsistent effects on visuospatial attention

**Andra Coldea** [1]*, **Stephanie Morand**[2], **Domenica Veniero**[3], **Monika Harvey**[1], **Gregor Thut**[1]*

**1** Centre for Cognitive Neuroimaging, Institute of Neuroscience and Psychology, University of Glasgow, Glasgow, United Kingdom, **2** School of Life Sciences, MVLS College, University of Glasgow, Glasgow, United Kingdom, **3** School of Psychology, University of Nottingham, Nottingham, United Kingdom

* gregor.thut@glasgow.ac.uk (GT); a.coldea.1@research.gla.ac.uk (AC)

## Abstract

Transcranial alternating current stimulation (tACS) is a popular technique that has been used for manipulating brain oscillations and inferring causality regarding the brain-behaviour relationship. Although it is a promising tool, the variability of tACS results has raised questions regarding the robustness and reproducibility of its effects. Building on recent research using tACS to modulate visuospatial attention, we here attempted to replicate findings of lateralized parietal tACS at alpha frequency to induce a change in attention bias away from the contra- towards the ipsilateral visual hemifield. 40 healthy participants underwent tACS in two separate sessions where either 10 Hz tACS or sham was applied via a high-density montage over the left parietal cortex at 1.5 mA for 20 min, while performance was assessed in an endogenous attention task. Task and tACS parameters were chosen to match those of previous studies reporting positive effects. Unlike these studies, we did not observe lateralized parietal alpha tACS to affect attention deployment or visual processing across the hemifields as compared to sham. Likewise, additional resting electroencephalography immediately offline to tACS did not reveal any notable effects on individual alpha power or frequency. Our study emphasizes the need for more replication studies and systematic investigations of the factors that drive tACS effects.

## Introduction

While the neural correlates of cognitive processes can be identified using brain imaging techniques, it is possible to obtain causal evidence on brain-behaviour relationships with the use of non-invasive (transcranial) brain stimulation methods. Transcranial alternating current stimulation (tACS), in particular, is of interest for probing causality between oscillatory activity of the brain and behaviour, as the sinusoidal tACS-currents hold promise to interact with intrinsic brain oscillations in a frequency-specific manner [1–4]. tACS has been gaining popularity in the last decade [3, 5], yet many controversies remain unresolved (see [6] for a review). For instance, it has been assumed that tACS-effects are caused by entrainment of brain oscillations and/or neuroplasticity [7–9]. However, concurrent recordings of electrophysiological data is

**Funding:** This work was supported by the ESRC (grant number ES/P000681/1 to A.C.). The funders had no role in study design, data collection and analysis, decision to publish, or preparation of the manuscript.

**Competing interests:** The authors have declared that no competing interests exist.

hindered by the presence of artefacts [10, 11], as a result of which the exact mechanisms of tACS-interaction with brain activity remain unclear. Likewise, it is unclear to what extent the low tACS-intensities that are in use can directly affect neuronal populations, given that much is being attenuated by the skin and skull [12–14], or alternatively exert their effects indirectly through transcutaneous co-stimulation of peripheral nerves [15]. Others have questioned to what extent these effects can be reproduced [16].

One domain that would seem ideal for testing the potential of tACS affecting performance through interacting with brain oscillations is visuospatial attention. Visuospatial attention refers to the ability of participants to allocate cognitive resources to a spatial location of interest, in order to prioritise and improve the processing of relevant stimuli at that position [17]. Numerous electro-/magnetoencephalography (EEG/MEG) studies have identified occipito-parietal alpha oscillations as correlates of visuospatial attention deployment, whereby alpha-power is suppressed contralaterally to the attended hemispace and/or enhanced contralaterally to the unattended position [18–25]. In addition, many EEG/MEG-studies have established a link between posterior alpha-power and specific behavioural outcomes in perceptual tasks, such as perceptual accuracy [20, 26–29] or subjective awareness of visual stimuli [30–35].

In the context of visuospatial attention, if occipito-parietal tACS at alpha-frequency were to bias behavioural performance in a spatially specific manner, this would be in (indirect) support of tACS causally interacting with underlying, perceptually relevant brain oscillations. Recently, Schuhmann and colleagues [36] have shown that applying high-density (HD) alpha-tACS over the left parietal cortex at 10Hz but not sham, induces a shift in visuospatial attention away from the contralateral right to the left hemifield. In analogy but adding concurrent EEG recordings, Kemmerer et al. [37] revealed that left parietal tACS at individual alpha frequency (IAF), but not at control frequencies (IAF±2 Hz) or sham, was associated with a left lateralization of alpha power, the magnitude of which predicted the right to leftward shift in visuospatial attention during endogenous shifts of attention. Similar results have been reported by Kasten and colleagues [38], who stimulated both the left and right occipital cortex with alpha- and gamma-tACS, while presenting participants with endogenous and exogenous visuospatial cues. A significant effect of tACS on endogenous but not exogenous attention was found when stimulation was applied over the left hemisphere, but not over the right [38]. Similarly, in the auditory domain, unihemi-spheric alpha-tACS caused a disruption in endogenous spatial attention contralaterally to the stimulated hemisphere [39, 40]. Together, these studies suggest that tACS can be used to establish a causal link between alpha oscillations and spatial attention, as well as highlight the potential of the technique to interact with brain oscillations and behaviour for potential clinical purposes, e.g. rehabilitation treatment of pathological asymmetries in visuospatial attention.

In the present study, we sought to replicate the significant behavioural effects of alpha-tACS on spatial attention, consistently reported in the literature so far (summarised in Table 1) to contribute to the evaluation of its efficacy and replicability to modulate spatial attention. Therefore, we designed our study in accordance with this literature. We largely followed the study protocol and design of Schuhmann and colleagues [36], including left parietal tACS at 10Hz using a high-density montage (central electrode at P3) with an assessment of the tACS-effects on spatial attention in the visual modality across the two visual fields (see Table 1). We tested a large sample of participants (n = 40, at the upper end of previous studies with positive findings, see Table 1) using the exact same task as Schuhmann et al. [36] measuring endogenous attention. We focused on task performance during tACS, as all previous studies reported consistent alpha-tACS effects on endogenous attention online to tACS (see Table 1). Finally, we applied tACS at 1.5mA for 20min (in the range of previous alpha-tACS studies with positive effects, see Table 1). We expected that with this design, that is 10 Hz tACS applied over the left posterior parietal cortex/P3, but not sham, we would induce a shift in attentional bias away

**Table 1. Summary of studies using alpha tACS to modulate spatial attention.**

| Study | Sensory modality tested | No. | Avg. age/ Age range/ Gender | α-tACS frequency | Montage | Area stimulated | Intensity applied | Duration of stimulation | Behavioural probe & effect direction |
|---|---|---|---|---|---|---|---|---|---|
| **Deng et al.** [39] | Auditory | N = 20 | 21.15; range 18–24; 13 F | 10 Hz vs sham | HD-tACS; central electrode: P2; return electrodes: CP2, P4, Pz, PO4. | R IPS | 1.5 mA | 20 min block | Online effects during tACS on endogenous attention; left hemispace affected |
| **Kasten et al.** [38] | Visual | N = 20 | 25±2.7; 10 F | IAF vs gamma (47 Hz) | Two pairs of circular electrodes: O1-P3 and O2-P4 | L and R occipital cortex | 2 mA | 8 min block | Online effects during tACS on endogenous (but not exogenous) attention in trials with invalid cues; leftward shift |
| **Kemmerer et al.** [37] | Visual | N = 21 | 45.38; range 19–72; 8 F | IAF vs IAF ±2 | HD-tACS; small circular electrode at P3 surrounded by a large ring electrode | L PPC | 1.5 mA | 35–40 min block | Online effects during tACS on endogenous attention (but not simple detection); leftward shift |
| **Schuhmann et al.** [36] | Visual | N = 36 | 21.56; range 18–29; 18 F | 10 Hz vs sham | HD-tACS: small circular electrode at P3 surrounded by a large ring electrode | L PPC | 1 mA | 35–40 min block | Online effects during tACS on endogenous attention (but not simple detection); leftward shift |
| **Wöstmann et al.** [40] | Auditory | N = 20 | Range 19–30; 10 F | 10 Hz vs sham | Round electrodes placed over FC5 and TP7 | L posterior STG, auditory and parietal regions | 1 mA | 25 min block | Online effects during tACS on endogenous attention; leftward shift |

Abbreviations: IPS, intraparietal sulcus; PPC, posterior parietal cortex; STG, superior temporal gyrus.

from the contralateral right to the left hemispace. Additionally, resting EEG was recorded immediately after stimulation to examine potential effects of tACS on individual alpha frequency and power.

## Materials and methods

### Participants

Forty-two healthy volunteers (mean age 22.4, range 19–38, 22 female) completed this study. An *a priori* sample size calculation based on the effect size observed in Schuhmann et al. [36] identified that a minimum of 38 participants were required for a repeated-measures ANOVA design ($d = 0.6$, $\alpha = 0.05$, power = 0.95). We therefore decided on a final sample size of 40 participants (pre-determined), but we had to record 42 as two participants were excluded from the final analysis, due to poor fixation during the experimental task, or noisy EEG recording, respectively. Participants gave informed written consent and had no contraindication to tACS (i.e. neurological/psychiatric disorders, history or family history of seizures or epileptic seizures, metal or medical implants, pregnancy, headaches, intake of central nervous system medication or recreational substances). All participants were naïve to tACS, reported normal or corrected-to-normal vision, and were right-handed according to the Edinburgh Handedness Inventory [41]. The procedures of the study were in line with the latest revision of the Declaration of Helsinki and were approved by the Ethics Committee of the College of Science and Engineering at the University of Glasgow.

### Procedure and task

Each participant underwent two sessions of maximally 1.5 hours each, at least 2 days apart. During these sessions, participants received active 10 Hz or sham tACS over the left parietal

cortex for 20 minutes (Fig 1A), while performing a visually cued target discrimination task (Fig 1B). The order of the two tACS sessions (10 Hz, sham) was counterbalanced across participants. Before the experiment, participants practiced one block of the behavioural task. The experimental task measured performance on endogenous attention (see Fig 1B, identical replication from [36]; stimulus material and script provided as a curtesy by these authors). Participants viewed stimuli on a computer screen (refresh rate, 60 frames/s) at a viewing distance of 57 cm. Each trial started with a fixation point presented for an interval ranging from 800 to 1200 ms, which turned into a bullseye for 500 ms. This was followed by a cue pointing either to the left (<< ● <<), right (>> ● >>), or both sides (<< ● >>), in anticipation of a forthcoming target. The cue was presented for 100ms and predicted with 80% accuracy the location of the target appearing after a 500 ms cue-target interval. The target stimulus was a Gabor patch tilted at 45˚ to either side (spatial frequency = 1.5 cycles per degree; envelope standard deviation = 0.75 degrees; Michelson contrast = 60%), appearing either in the left or right hemifield at 7˚ eccentricity (Fig 1B) and presented for 100 ms. Participants had to discriminate whether the Gabor patch was oriented clockwise or counterclockwise and were instructed to respond as fast and as accurately as possible once the target appeared on the screen, by pressing the left and right arrow keys on the keyboard, using the index and middle finger of their right hand, respectively. They were instructed to keep their eyes on the fixation point throughout the trial. The next trial started immediately after a response was made. One experimental session consisted of 336 trials containing 192 valid trials (i.e. target was presented in the cued hemifield), 48 invalid trials (i.e. target was presented opposite the cued hemifield), and 96 neutral trials (i.e. target was preceded by a neutral cue). The task duration was approximately 20 minutes, with self-paced breaks every 84 trials.

After task completion and tACS cessation, 4 minutes of resting EEG was recorded from three occipital electrodes to evaluate the amplitude and individual peak frequency in the alpha band (8-12Hz) across conditions (tACS and sham). At the end of each session, a questionnaire was administered to assess how well the participants tolerated the tACS stimulation. Furthermore, to assess whether participants were blinded to the stimulation protocol, an additional questionnaire was administered at the end of the second session, in which participants had to judge in which session they received real stimulation and in which session sham.

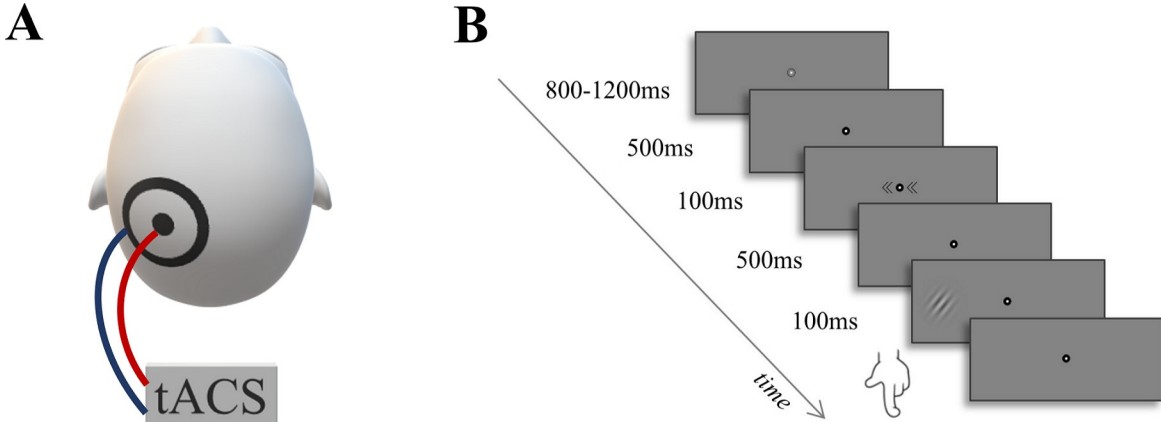

**Fig 1. Experimental setup. A.** tACS setup. A small circular electrode was positioned over P3 and a large electrode was centring it. Figure adapted from Schuhmann et al. [36] **B.** Stimulus schematics and trial time course. The trial started with the presentation of a fixation point, followed by a cue (here: left). The target stimulus was a sinusoidal grating tilted at 45˚ to either left or right, presented on either side of the screen (here: presented right). Participants had to indicate the direction in which the grating was tilted. Figure adapted from Schuhmann et al. [36].

## Transcranial alternating current stimulation

High-density tACS was delivered through a battery-driven, constant current stimulator (NeuroConn GmbH, Ilmenau, Germany) using a rubber ring tACS electrode with a small central, circular electrode (2.1 cm diameter, 3.5 cm$^2$; thickness: 2 mm) and a large outer ring (9 cm inner and 11 cm outer diameter, 31.5 cm$^2$; thickness: 2 mm) (as in [36]). This montage was chosen to ensure a high spatial focality [42]. The small circular electrode was positioned in accordance with the International 10–20 EEG montage over the left parietal cortex (P3), with the large electrode surrounding it (Fig 1A, again as in [36]). The electrodes were applied on the scalp using conductive gel (ten20 paste, Weaver and Company, Aurora, CO, USA). Electrode impedance was kept below 10 kΩ. Stimulation frequency was 10Hz (as in [36]) but the intensity was set slightly higher to 1.5 mA (peak-to-peak), yielding an average current density of 0.4 mA/cm$^2$ at the central electrode, and 0.05 mA/cm$^2$ at the surround electrode. For a picture with the simulated voltage distribution, we refer to Schuhmann et al. [36], their Fig 1A. tACS was administered in a within-subject design with one active condition and one sham condition. In the active condition, phase offset was set to 0 at the start and 100 cycles were used for ramping up, with the stimulator being switched off after completion of the experimental task. The stimulation duration was approximately 20 minutes. In the sham condition, the stimulator was ramped up and then immediately ramped down, each within 100 cycles.

## Eye tracker

Eye tracking (Eyelink 1000, SR Research, Mississauga, Ontario, Canada) was used during the experimental task to ensure fixation before stimulus presentation. A 9-point calibration and validation procedure was carried out before the start of the experimental task and then again prior to the start of each of the four blocks of trials. Data were acquired using monocular tracking of the right eye at a sampling rate of 1000 Hz.

## Electrophysiological data recording

Immediately after completion of the experimental task and the tACS stimulation, Ag/AgCl electrodes were attached to the scalp of participants using conductive gel (ten20 paste, Weaver and Company, Aurora, CO, USA). A small number of electrodes was chosen to minimize the gap between end of tACS and start of EEG recording (~5min). Resting EEG was then recorded for a total of 4 minutes (2 minutes eyes closed; 2 minutes eyes open) from the occipital sites O1, Oz, and O2 (referenced to AFz), according to the international 10–20 Electrode Montage, using a BrainAmp MRPlus amplifier (BrainProducts GmbH, Munich, Germany). Electrode impedance was kept below 10 kΩ and EEG data were acquired at a sampling rate of 1000 Hz.

## Data analysis

**Behavioural analysis.** Pre-processing of the behavioural data was conducted in Matlab (MathWorks, Natick/USA). Following the procedure of Schuhmann et al. [36], trials were removed *post-hoc* if the eye movements during a trial exceeded 2˚ of visual angle in the time window starting 100 ms before the cue until stimulus onset. On average, 1.7% of all trials were discarded per participant due to eye movements. Trials were also excluded if the reaction times (RTs) were extreme (i.e. < 120 ms, > 800 ms). For the analysis of reaction times, only correct trials were included.

For each participant, accuracy and median RTs were computed for each tACS condition (i.e. 10Hz tACS vs sham), type of cue (i.e. invalid, neutral, valid) and target location (i.e. left hemifield vs right hemifield), in analogy to Schuhmann et al. [36]. Because the RT

distributions are usually skewed [43], we also conducted the analyses using the log-transformed data. The results of the analyses remained qualitatively unchanged, not affecting the conclusions, hence these analyses are not reported in the paper. Spatial bias was calculated by subtracting the RT/accuracy in the right hemifield from the RT/accuracy in the left hemifield ($RT/Accuracy_{Left\ hemifield} - RT/Accuracy_{Right\ hemifield}$).

**EEG analysis.** The EEG analysis was conducted in BrainVision Analyzer 2.0 (Brain Products) using a semi-automated approach. The post-tACS continuous EEG signal for both resting "eyes closed" and "eyes open" was segmented into 1 s epochs. A fast Fourier transform (FFT) was calculated for frequencies between 0.1 and 50 Hz using a Hanning window. For each participant, the resulting spectra of each tACS session were averaged across epochs. The frequency window for the analysis of the data was set between 8 and 12 Hz, within which the IAF peak and corresponding amplitude were identified.

**Statistical analyses.** Statistical analyses were performed using R 3.4.1 [44]. To ensure that the attention manipulation was effective, we first performed a repeated measures analysis of variance (rm-ANOVA) with the within-subject factor cue validity (invalid, neutral, valid) on the median RT of the sham data only (with the data collapsed across the target locations). To verify the presence of a hemifield/ attentional bias as reported by Schuhmann et al. [36] (RT left > right visual field), we also ran a rm-ANOVA with the within-subject factor hemifield (left, right) on the median RT of the sham data. The main analyses then followed the same steps as Schuhmann and colleagues [36] and consisted of a rm-ANOVA with the factors tACS condition (10Hz, sham), and cue validity (invalid, neutral, valid) on the hemifield bias (median $RT_{Left\ hemifield} -$ median $RT_{Right\ hemifield}$). When sphericity was violated, Greenhouse-Geisser corrected values are reported. Where appropriate, t-statistics were employed to test simple effects.

Given the null results (see below), several additional exploratory analyses were run including on accuracy and using analyses of covariance (ANCOVA) to explore whether the effects of tACS may depend on specific individual (trait) factors. The ANCOVA analyses mirrored the main rm-ANOVA, such that two within-participant factors were included: tACS condition (10Hz, sham) and cue validity (invalid, neutral, valid), in addition to the covariates. We explored the influence of the following four covariates on tACS outcome (in four different ANCOVAs): an individual hemifield bias, IAF, deviation of IAF from 10 Hz (absolute difference), and alpha power; all inferred during the sham session to reflect individual trait factors unaffected by tACS. Because of our within-subjects design, covariates have been centred by subtracting the average covariate value from each covariate score, to increase the precision of the analyses [45]. A significant effect of the covariate on tACS outcome would be reflected in a significant interaction either between the covariate and tACS condition and/or a significant triple interaction between the covariate, tACS condition, and cue validity. Additionally, we also analysed potential effects of tACS on resting EEG and peripheral sensations.

## Results

### Main analyses

**RTs.** We first checked whether the experimental manipulation of spatial attention was effective by analysing RTs in the sham condition only. This was confirmed by a repeated measures ANOVA on the median RTs (data averaged across hemifields, Fig 2A) revealing a significant main effect of cue validity ($F(2,78) = 39.9$, $p < .001$, $\eta^2_G = .03$). Responses in valid trials (M ±SD: 450.7±70.5ms) were significantly faster than in neutral trials (464±76ms; $t(39) = -5.2$, $p < .001$, $r^2 = .38$, Bonferroni corrected), and faster than in invalid trials (485.7±85.8ms; $t(39) = -7.2$, $p < .001$, $r^2 = .49$, Bonferroni corrected), while responses in neutral trials were significantly faster than in invalid trials ($t(39) = 5.22$, $p < .001$, $r^2 = .38$, Bonferroni corrected).

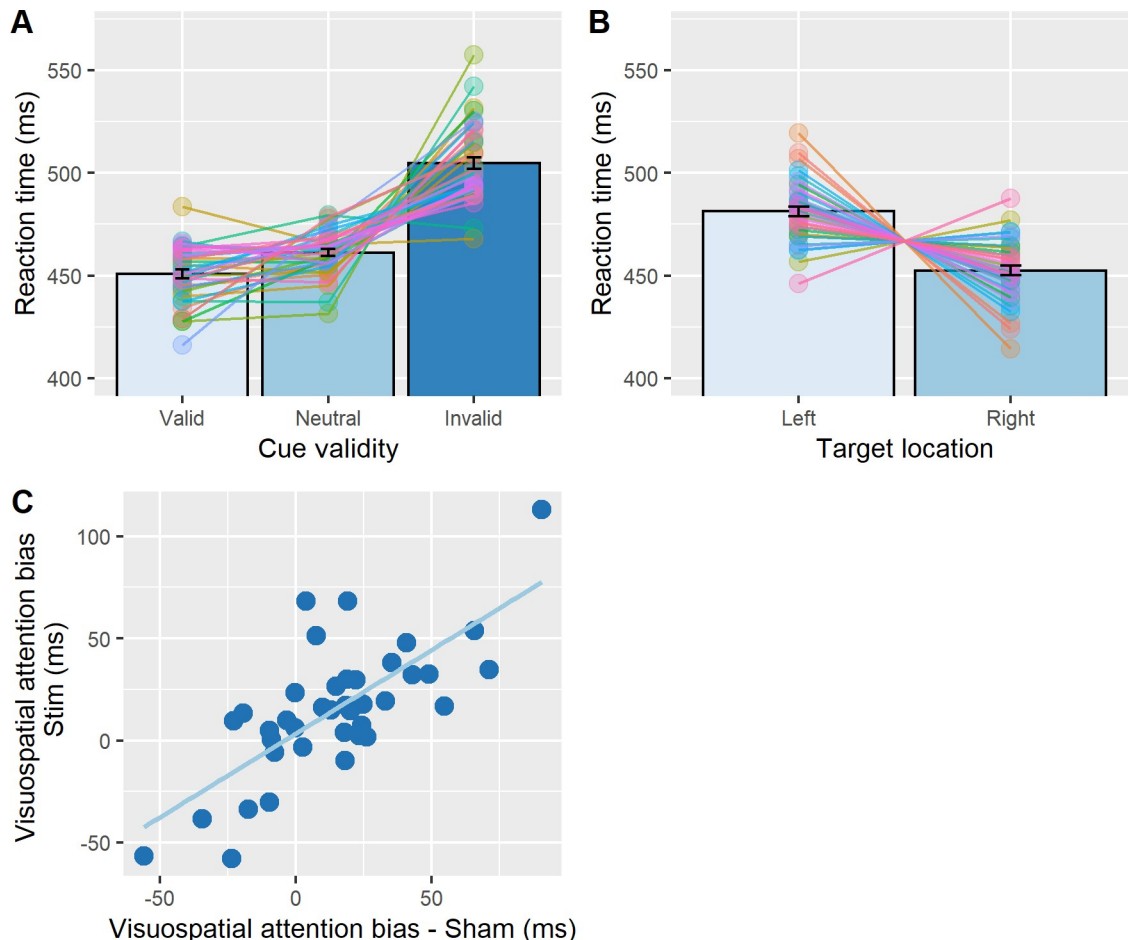

**Fig 2. Experimental checks. A. Cueing effect:** RTs were averaged across target location for each type of cue (sham session only). Significantly faster RTs were found for valid trials, as compared to neutral and invalid trials. RT in neutral trials were significantly faster than invalid trials. Error bars represent 95% confidence interval corrected for a within subjects design [46]. The bar plots have been superimposed with individual data points. **B. Hemifield bias:** RTs were averaged across cue validity conditions for each target location (sham session only). Significantly faster RTs were found for trials in which the stimuli were presented in the right hemifield, as compared to the left. Error bars represent 95% confidence interval corrected for a within subjects design [46]. Similarly, the bar plots have been superimposed with individual data points. **C. Correlation of measure of hemifield bias between the two experimental sessions**. Since the intercept is close to 0 (i.e. 3.3 ms) and the slope is close to 1 (i.e. 0.8), the model already indicates that the spatial bias in the two experimental sessions ($RT_{\text{Left hemifield}} - RT_{\text{Right hemifield}}$) is very similar and therefore a significant effect of stimulation is unlikely.

We then tested whether there was a difference between the RTs in the left as compared to the right hemifield in the sham condition (RT left > right visual field), as reported by Schuhmann et al. [36] employing the same paradigm. A t-test on median RTs (data averaged across cue validity, Fig 2B) indeed revealed a significant difference between hemifields ($t(39) = 3.13$, $p = .003$, $r^2 = .24$). Participants responded significantly faster when stimuli were presented in the right visual field ($459.6 \pm 81.3$ms) than the left visual field ($474 \pm 75.4$ms), replicating Schuhmann et al. [36]. This result suggests that on average, participants had a rightward bias overall.

Before testing the main hypothesis that left parietal alpha-tACS but not sham affects this rightward bias, we wanted to check how consistent this measure of bias was within participants. To this end, a Pearson's product-moment correlation coefficient was calculated for the bias measures obtained in each session. There was a significant positive correlation between

the rightward bias during stimulation versus sham (r = .73, p < .001; see Fig 2C), suggesting that this is a reliable, within-participant trait measure.

Our main analysis then examined whether left parietal tACS induced a bias away from the right to the left hemifield when applied at 10Hz as compared to sham, possibly as a function of cue condition (as reported by [36], see also Table 1). To this end, we ran a repeated measures ANOVA with the factors tACS condition (i.e. 10 Hz tACS, sham) and cue validity (i.e. invalid, neutral, valid) on the spatial bias measure ($RT_{Left\ hemifield} - RT_{Right\ hemifield}$) (see Fig 3A for the corresponding data). There was no significant main effect of tACS condition (F(1, 39) = .04, p = .83, $\eta_G^2$ = .0001) and no significant interaction with cue validity (F(1, 78) = .52, p = .55, $\eta_G^2$ = .001). These results show that left parietal tACS did not shift the bias to the left, as compared to sham, irrespective of cueing condition. However, we found a significant main effect of cue validity (F(1, 78) = 5.78, p = .01, $\eta_G^2$ = .02). Averaged across stimulation conditions, there was a

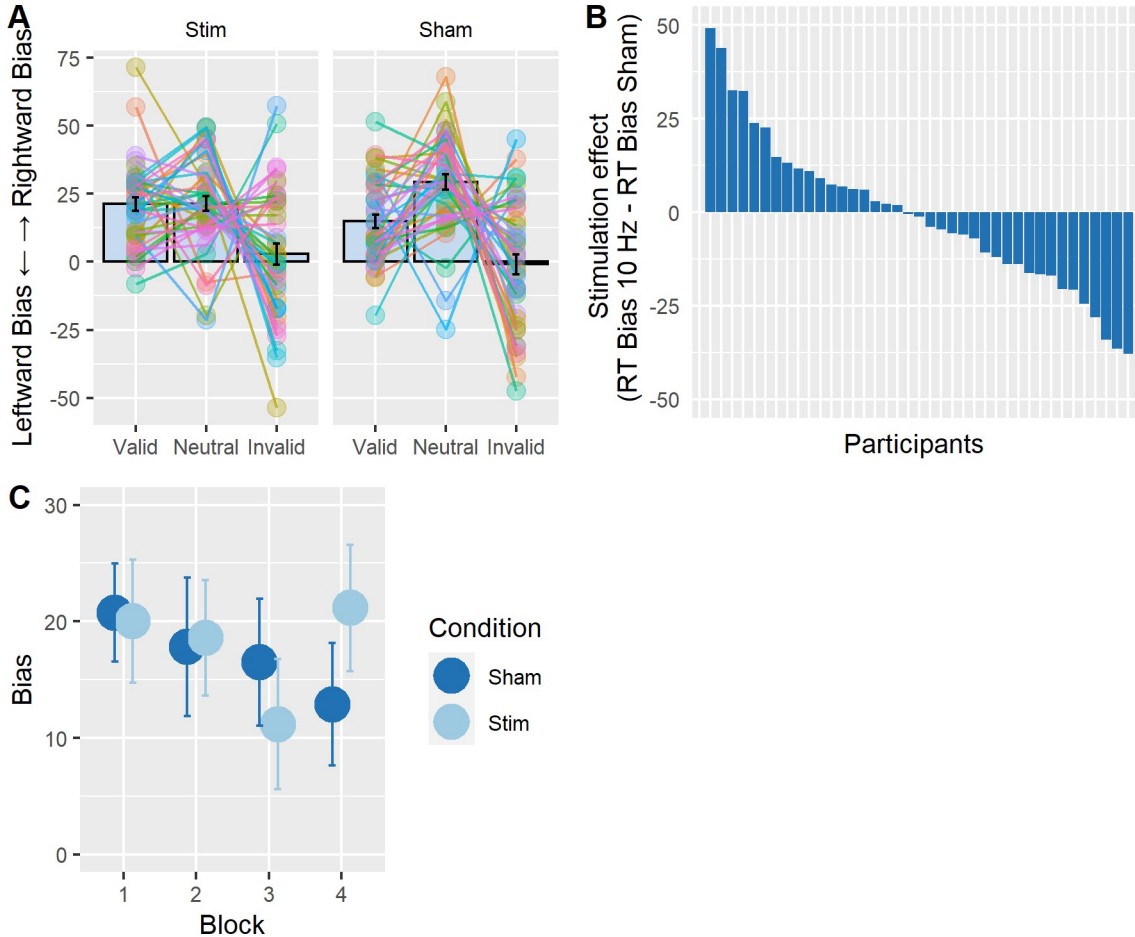

**Fig 3.** No tACS effects on hemifield bias **A.** Measure of spatial bias across simulation and validity conditions ($RT_{Left\ hemifield} - RT_{Right\ hemifield}$). A positive value indicates a rightward bias (i.e. faster RTs in the right hemifield), whereas a negative value indicates a leftward bias (i.e. faster RTs in the left hemifield). The average values for each condition are superimposed with individual data points of each participant. Error bars represent 95% confidence interval corrected for a within subjects design [46]. **B.** Stimulation effect per participant on spatial bias ($RTBias_{10Hz\ tACS} - RTBias_{sham}$). A negative value means that participants had a greater leftward (more negative) spatial bias with 10 Hz stimulation as compared to sham (expected direction). **C.** Change in the measure of spatial bias across the 4 experimental blocks (~5 min). The plot displays the average spatial bias per block and the lines represent the standard error, where a positive value of bias indicates a rightward bias. There was no significant difference between the stimulation conditions with time-on-task.

greater rightward bias for neutrally cued trials (20±36.2ms) than valid (16.4±31.4ms) and invalid trials (7.8±40.2ms). Additionally, when computing the average spatial bias change per participant across session (RT Bias$_{10Hz\,tACS}$−RT Bias$_{sham}$), we found that around 50% of all participants (n = 21 out of 40) showed a greater leftward bias in the 10 Hz tACS condition, compared to sham (Fig 3B), which would be expected by chance. Given these null results obtained by employing the same analysis as Schuhmann et al. [36], we ran several exploratory analyses reported below.

## Exploratory analyses

**Effect of stimulation on spatial bias (RT) across time.**   We first checked whether the effects of tACS on the spatial bias as measured by RT may have occurred only towards the end of the 20 min stimulation session. To this end, the data were split into blocks of ~5 min each (4 blocks of 84 trials) and average RTs were re-calculated for each participant and condition. Trials had to be collapsed across validity conditions, because there was an insufficient number of invalid trials to allow calculation of the spatial bias measure per block. A repeated measures ANOVA with the factors tACS condition (i.e. 10 Hz tACS, sham) and block (i.e. 1, 2, 3, 4) on the spatial bias measure (RT$_{Left\,hemifield}$−RT$_{Right\,hemifield}$) (see Fig 3C) revealed no significant main effect of tACS condition (F(1, 39) = .6, p = .8, $\eta_G^2$ = .0001), nor block (F(3,117) = 1.31, p = .27, $\eta_G^2$ = .005), and no significant interaction (F(3, 117) = 1.94, p = .12, $\eta_G^2$ = .005), which suggests that participants maintained a consistent level of spatial bias throughout the experiment for both stimulation conditions. Upon visual inspection, a difference between the two stimulation conditions seemed to appear in the last 5 minutes of stimulation, yet a t-test on the spatial bias during 10 Hz versus sham in block 4 was not significant (t(39) = -1.82, p = .07 $r^2$ = .14). Please also note that the observed pattern would be against the predictions (more rightward bias with left parietal tACS compared to sham).

**Dependency of tACS-effects (RT) on trait factors: Individual spatial bias and alpha-frequency/power.**   As previous studies using transcranial electrical stimulation have indicated that the effects may depend on the brain state and individual trait factors [47, 48], we explored whether tACS outcome in our study may have depended on four such factors.

First of all, we re-analysed the RT data as a function of the individual (trait) bias in visuospatial processing, that we estimated from the sham data. To this end, we ran an ANCOVA mirroring the main rm-ANOVA analysis, with the factors tACS condition and cue validity on the dependent measure of hemifield bias, adding individual bias as a covariate. After controlling for the individual bias, the ANCOVA revealed a significant main effect of cue validity (F(2,76) = 5.722, p = .005) as before. However, the interaction between the covariate and tACS condition was not significant (F(1,38) = 2.127, p = .153), nor was the triple interaction between the covariate, tACS condition and cue validity (F(2,76) = .912, p = .406), suggesting that the directionality of the individual bias as measured in the sham session did not impact the effect of tACS stimulation on the hemifield bias.

Next, we wanted to investigate whether tACS outcome depended on participants' brain oscillations as recorded in the sham session (based on the eye-closed data from the left electrode O1, see EEG below). To test this, we ran three ANCOVAs as above but with the covariates individual alpha frequency (IAF), deviation of IAF from the 10Hz stimulation frequency (absolute difference), and alpha power. Interactions of tACS with underlying brain oscillations may be enhanced if tACS frequency (here 10Hz) matches IAF (e.g. [7, 9]) Additionally, previous studies have reported effects of alpha tACS to depend on alpha power at baseline (e.g. [47]). There was a significant main effect of cue validity in all these analyses (p < .05), but no significant interactions were found in these analyses (interaction between the covariate and

tACS condition: IAF F(1,38) = .654, p = .423, deviation of IAF from 10 Hz F(1,38) = .023, p = .878, alpha power F(1,38) = .383, p = .539; all triple interactions between the covariate, tACS condition and cue validity: IAF F(2,76) = .8, p = .45, deviation of IAF from 10 Hz F(2, 76) = .159, p = .85, alpha power F(2,76) = .809, p = .448). This indicates that the stimulation effect was not impacted by individual alpha frequency and/or alpha power.

We note though that our exploratory analyses of the impact of covariates was post-hoc, and our design not optimal for inferring individual trait factors, as inferred during sham (counterbalanced with tACS), when these should have ideally been inferred before any experimental manipulation.

**Effects of tACS on EEG.** Resting EEG was recorded closely after tACS with both eyes open and eyes closed. Using the data recorded from O1, the test-retest reliability for identifying IAF was probed. A Pearson product-moment correlation coefficient was computed to assess the relationship between $IAF_{10Hz\ tACS}$ and $IAF_{Sham}$, revealing a weak positive correlation between the two variables in the eyes open condition (r = .32, p = .03), and a stronger correlation in the eyes closed condition (r = .93, p < .001, Fig 4A). Equivalent results were observed for alpha power, where a weak positive correlation was found between $\alpha$-power$_{10Hz\ tACS}$ and $\alpha$-power$_{Sham}$ during eyes open (r = .47, p = .001) and a stronger correlation during eyes closed (r = .91, p < .001, see Fig 4D). Due to the better test-retest reliability (SNR) of both IAF and power during eyes closed, we proceeded with the EEG analyses of the eyes-closed data only.

*tACS-effects on alpha-frequency.* To test whether tACS aligned IAF to the stimulation frequency, which would be in accordance with an entrainment effect of tACS [1, 3, 49], we ran a t-test on the difference IAF minus 10Hz (absolute difference) between 10Hz tACS and sham on data recorded from electrode O1, i.e. ipsilateral to the stimulation site. If entrainment occurred, the IAF of the participants should be closer to 10 Hz following active stimulation as compared to sham. No significant difference was found between the two conditions (t(39) = -1.93, p = .06, $r^2$ = .15, Fig 4B). We also compared IAF peaks during the two tACS sessions (again using a t-test on the recordings from electrode O1) and found a significant difference between 10 Hz tACS and sham (t(39) = -3.83, p < .001, $r^2$ = .28, Fig 4C). Similar results of small effect size were found for data recorded from electrode O2 (i.e. contralateral to the stimulation site) (t(39) = -2.29, p = .02, $r^2$ = .17). Note that this significant tACS effect on IAF was very small in magnitude (an increase of 0.185Hz; from 9.98Hz for sham to 10.165Hz for alpha-tACS), and unexpected/unexplained, and is therefore not further discussed.

*tACS-effects on alpha-power.* Equivalent analyses were conducted on alpha power. T-test revealed no significant differences in power between sham and 10 Hz tACS, neither for electrode O1 (t(39) = -.06, p = .95, $r^2$ = .004, Fig 4E) nor O2 (t(39) = -.73, p = .46, $r^2$ = .05).

**Accuracy.** Our main analysis focused on RT, as this measure was shown to be affected by tACS in Schuhmann et al. [36]. Although the overall accuracy was 95% in our participants (ranging from 73% to 100%) and hence close to ceiling (cf to 93% in [36]), we also checked for potential tACS effects on this measure. A repeated measures ANOVA on the median accuracy in the sham condition (data averaged across hemifields) revealed a main effect of cue validity (F(2,78) = 3.34, p = .04, $\eta_G^2$ = .02). Participants were significantly more accurate in valid trials (95.9±4%) than in invalid trials (94.5±5.7%) (t(39) = 2.33, p = .02, $r^2$ = .18). There was no significant difference in accuracy between valid and neutral trials (95.1±4.7%) (t(39) = 1.73, p = .09, $r^2$ = .13), nor between neutral and invalid trials (t(39) = -1.06, p = .3, $r^2$ = .08).

We also tested whether accuracy differed between the two hemifields during the sham condition but found no effect. The repeated measures ANOVA on median accuracy (data averaged across cue validity) was not significant (F(1,39) = .34, p = .56, $\eta_G^2$ = .002), indicating that participants' accuracy was consistent regardless of stimulus location.

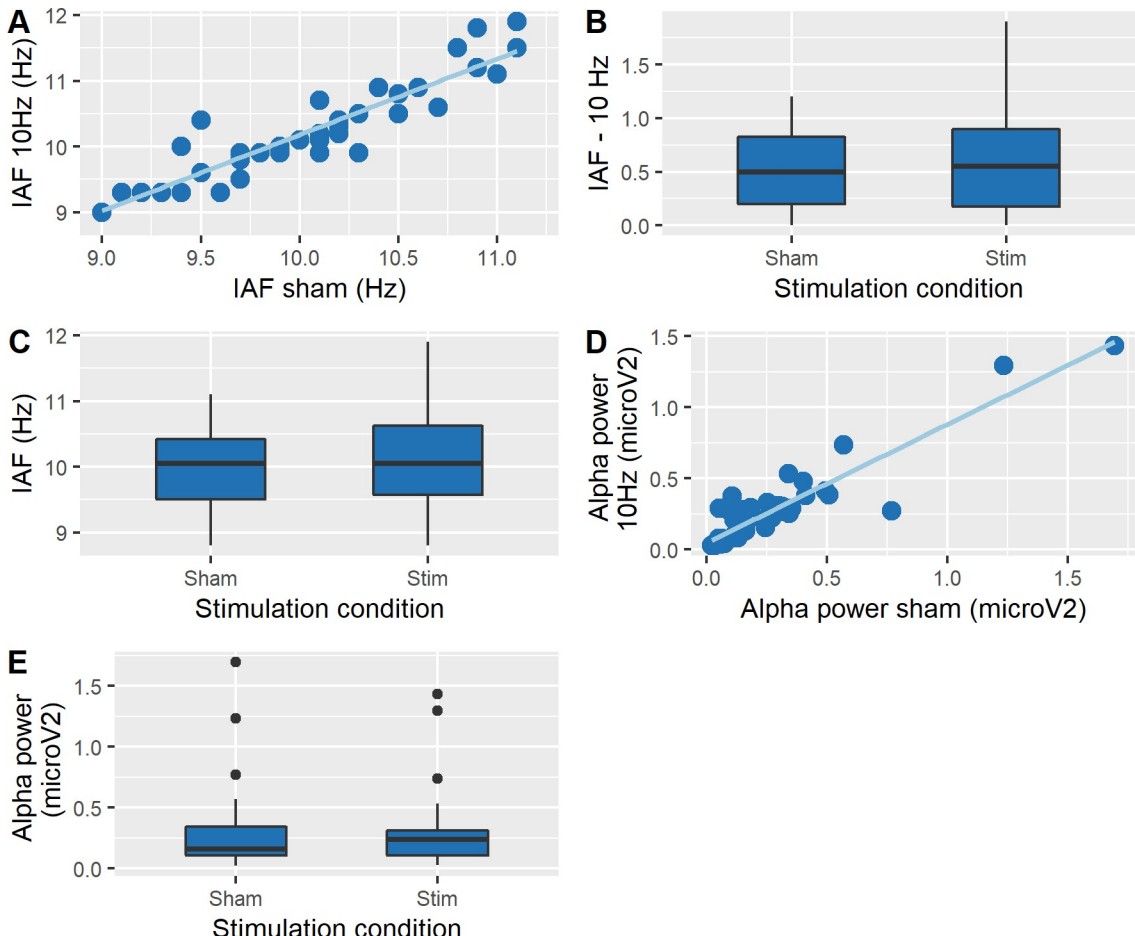

**Fig 4. No tACS effects on EEG (eyes closed data). A.** Relationship between $IAF_{Stimulation}$ and $IAF_{Sham}$ showing a good test-retest reliability. **B.** Absolute difference between IAF and 10 Hz during sham and stimulation. There was no significant difference between the two stimulation conditions, indicating there is no evidence for entrainment in our sample (convergence of IAF to 10Hz tACS frequency = zero after tACS relative to sham). **C.** IAF during sham and stimulation. IAF was slightly (by 0.185 Hz) but significantly increased after tACS relative to sham. **D.** Relationship between alpha-power$_{Stimulation}$ and alpha-power$_{Sham}$. **E.** Alpha power during sham versus stimulation, with no significant difference between the two conditions. All data shown are from electrode O1. The boxplots show a representation of the median and the first and third quartiles. The whiskers of the boxplot can take a maximal value up to 1.5*interquartile range, with all the values exceeding the whiskers being outliers.

Next, we calculated a measure of hemifield bias based on accuracy ($Accuracy_{Left\ hemifield}-Accuracy_{Right\ hemifield}$) per condition (Fig 5) and examined potential effects of tACS (10 Hz tACS, sham) on this bias measure as a function of cue validity (i.e. invalid, neutral, valid), using a repeated measures ANOVA. In analogy to the analysis on RTs, this did not reveal any significant main effect of tACS condition ($F(1,39) = 0.907$, $p = .346$, $\eta_G^2 = 0.003$), nor an interaction with cue validity ($F(2, 78) = .336$, $p = .70$, $\eta_G^2 = .001$), and also no main effect of type of cue ($F(2,78) = 1.272$, $p = .28$, $\eta_G^2 = .009$).

**Effects of tACS on sensations and blinding.** After each of the sessions, participants were asked to fill in a questionnaire regarding how well tACS was tolerated. A total of seven different sensations were rated on a scale from 1 (no sensations felt during the experiment) to 5 (strong sensations felt during the experiment). Table 2 shows the average ratings for each of these sensations. At the end of the second session, participants were asked to report in which session they thought they received real stimulation and in which sham. Of the 40

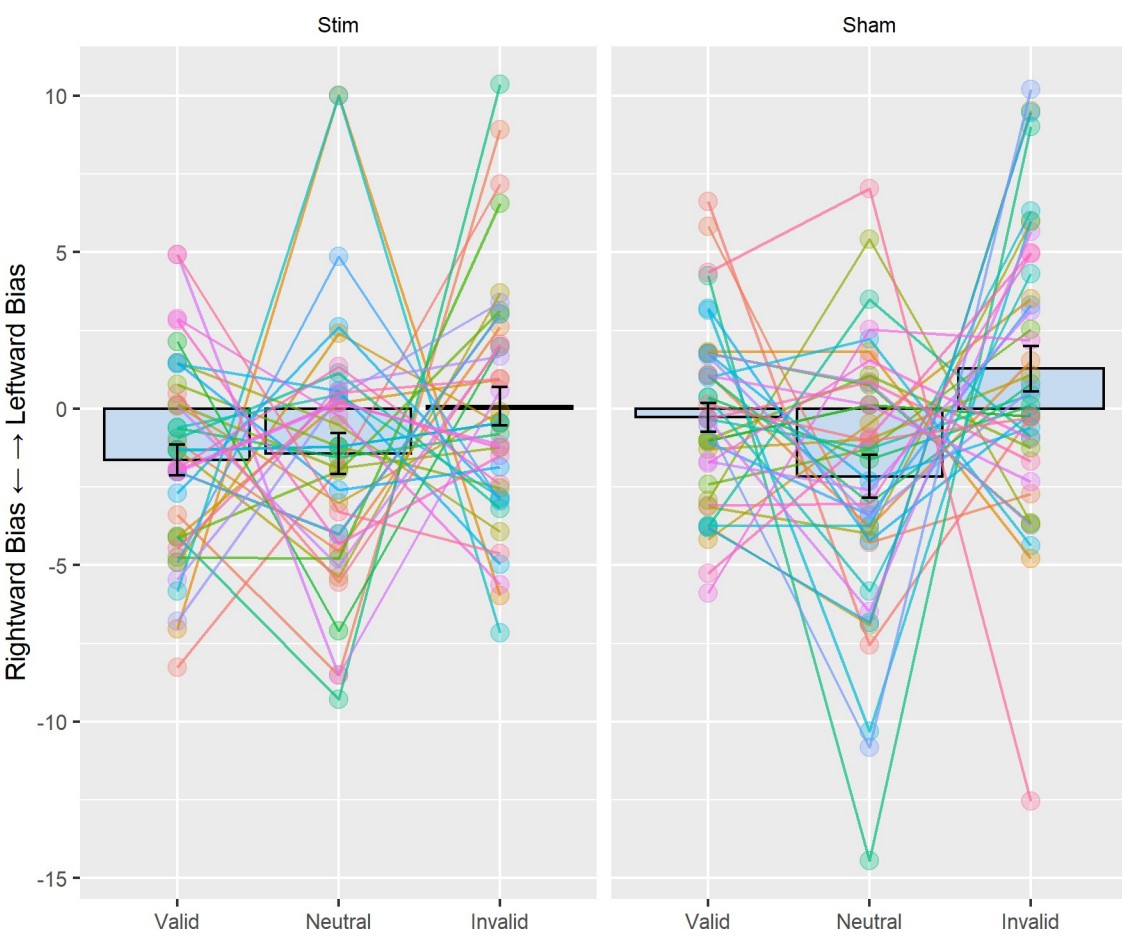

**Fig 5. Accuracy.** The average accuracy for each condition superimposed with individual data points of each participant. Error bars represent 95% confidence interval corrected for a within subjects design [46]. A positive value of the measure of bias in accuracy indicates a leftward bias (i.e. increased accuracy in the left hemifield), and a negative value indicates a rightward bias (i.e. increased accuracy in the right hemifield).

participants whose data was included in the analysis, 25 were able to correctly distinguish between 10 Hz tACS and sham sessions. A chi square goodness of fit performed to compare the percentage of correct guesses (62.5% = 25/40) with the expected occurrence by chance (50%: 20/40) revealed no significant deviation from the expected value ($X^2(1) = 2.5$; $p = 0.12$), thus confirming that the percentage of participants correctly identifying the sham condition was not different from chance.

We then tested whether there was an effect of 10 Hz tACS vs sham on the sensations reported by the participants. There were no significant differences between the two experimental conditions on the intensity of any of the seven sensations reported. The largest

**Table 2. Average intensity of the sensations felt during the experiment as reported by participants on a scale from 1 (no sensations) to 5 (strong sensations).**

| Stimulation Condition | Itchiness | Pain | Burning | Warmth/ Heat | Pinching | Iron taste | Fatigue |
|---|---|---|---|---|---|---|---|
| 10 Hz | 1.43 | 1.28 | 1.3 | 1.48 | 1.45 | 1 | 1.43 |
| Sham | 1.3 | 1.23 | 1.15 | 1.4 | 1.33 | 1.05 | 1.53 |

difference was found in the reports regarding burning (t(39) = -1.43, p = .16, uncorrected) and iron taste (t(39) = 1.43, p = .16, uncorrected).

## Discussion

Several recent studies using alpha tACS have reported consistent effects on behavioural measures of spatial attention in both the visual and auditory modalities during tACS [36–40]. Here, we tested this effect using the same endogenous attention task, stimulation site and high-density tACS setup as Schuhmann and colleagues [36], as well as a tACS intensity and duration of similar magnitude than other groups reporting effects [37–40]. Based on this prior literature, we expected that 10 Hz tACS applied over the left posterior parietal cortex should induce a shift in bias away from the right and towards the left hemispace. In contrast with this prior literature, we failed to find a tACS effect using our parameter combinations, as there was no significant difference between 10 Hz tACS and sham.

Taking into account the evidence coming from EEG [18, 20, 23, 25] and the tACS literature [36–40] supporting a role of alpha oscillations in visuospatial attention, our results are unexpected. In our design, we got closest to the study of Schumann et al. [36], implementing the exact same protocol, except for using higher intensity (1.5 mA instead of 1 mA) but shorter stimulation duration (20 min instead of 40 min). Our null results could therefore be attributed to the change in parameters that were implemented here as compared to Schuhmann and colleagues [36]. However, the efficacy of alpha tACS to shift spatial bias away from the contra- to the ipsilateral hemispace has been shown for a range of stimulation intensities (1–2 mA) and durations (8–40 minutes; see Table 1 for more details), suggesting that intensity and duration are poor predictors of outcomes of alpha-tACS on spatial perceptual bias. Our results are in line with other negative findings reported in the tACS literature. Hopfinger and colleagues [50] showed that 10 Hz tACS had no effect on endogenous attention, although tACS was applied to the right not the left hemisphere. Similarly, Veniero and colleagues [51] assessed the effect of right hemispheric alpha tACS on visuospatial attention, using a variant of the line bisection task. While their initial experiment yielded statistically significant effects of tACS, the results were not confirmed in a subsequent internal replication [51]. Even though we could not replicate the behavioural effect of tACS on task performance, we have conducted further exploratory analyses to determine whether the effect of tACS depended on the brain state and individual trait factors of the participants, namely the individual spatial bias, IAF, deviation from IAF, and alpha power, as recorded from the sham session. Although in the literature it is reported that the outcome of brain stimulation techniques is state/trait-dependent (see also [47, 48, 52, 53]), we were unable to provide supportive evidence for such a dependency of alpha tACS effects for our dependent measure. However, our analyses was post-hoc and exploratory so further evidence is needed to better understand the effects of these covariates on the effect of tACS as measured here.

An inconsistent picture also emerges when examining studies attempting to use transcranial direct current stimulation (tDCS) to shift attention bias. In an experimental paradigm similar to Schuhmann et al. [36], Duecker et al. [54] tested whether parietal tDCS could be used to induce an interhemispheric imbalance that would shift attention away from the right towards the left hemifield. They attempted to decrease cortical excitability through cathodal tDCS over the left hemisphere, while increasing cortical excitability with anodal tDCS over the right. No effect of bihemispheric tDCS was found on the attentional bias, although it was reported that stimulation led to an impairment of attentional benefits (i.e. faster reaction times for trials when the cue was valid as opposed to neutral) in the right hemifield for endogenous orienting [54]. Similarly, Li and colleagues [55] used oppositional parietal tDCS in a modified Posner task but found no effect of stimulation on spatial attention. However, shifts in visual

attention following tDCS stimulation have been reported in perceptual line bisection paradigms [56, 57]; but see [51], as well as for visual localisation [58], where a left-anodal right-cathodal montage has induced a rightward bias.

Here, we demonstrate variable effects of tACS when targeting alpha oscillations for the purpose of modulating visuospatial attention in healthy participants. However, tACS has been successfully used for modulating alpha and beta oscillations in relation to other visual processes, such as temporal [59–63] and spatial binding [64]. Our study and negative results should hence not be taken to generalize to other relationships between brain oscillations and perceptual processes and their tests through non-invasive brain stimulation techniques. Also, our study in healthy participants may not be generalizable to patients. Alpha-tACS could be clinically relevant [65–67], for example in patients who have suffered right hemispheric damage following stroke and show attentional impairments (known as neglect; [68]). One limitation of our design in regards to a clinical implementation is the single session protocol. Recent experiments employing multi-session designs [69–71] and/or stimulating at the individualized alpha frequency [72] have demonstrated the potential of tACS as a therapeutic intervention for psychiatric disorders. The lack of these manipulations in our and previous studies on spatial attention may explain some of the observed variability.

A survey on research practices targeting neuroscientists employing transcranial electrical stimulation techniques reported that only 45–50% of respondents were able to routinely replicate published effects [16], although concerns regarding reproducibility have been extended to the whole scientific community [73–75]. In recent years, the tACS literature has seen a surge in studies reporting null effects [76–81] and failed replications [51, 77, 82–84]. This calls for a more systematic investigation of the factors that are driving these inconsistencies. In our study, although coming close to Schuhmann and colleagues [36], we did unfortunately not fully mirror their design, hence inferences regarding the (in)effectivity of a particular parameter combination for shifting spatial attention are elusive. More direct replication studies of effects reported in the literature to better characterize the factors that determine the efficacy of tACS are needed.

## Acknowledgments

We would like to thank Teresa Schuhmann, Selma K. Kemmerer, Felix Duecker, Tom A. de Graaf, Sanne ten Oever, Peter De Weerd, and Alexander T. Sack, for their generosity, providing us with the code for the task and invaluable advice in setting up the experiment and analysing the data.

We would also like to thank Heather Brown, Richard Hohne, Olivia Gill, Xiya Lin, Alex Manterfield and Richard Hammond for their contribution to the project.

## Author Contributions

**Conceptualization:** Andra Coldea, Stephanie Morand, Domenica Veniero, Monika Harvey, Gregor Thut.

**Data curation:** Andra Coldea, Stephanie Morand, Domenica Veniero.

**Formal analysis:** Andra Coldea, Gregor Thut.

**Software:** Andra Coldea.

**Supervision:** Monika Harvey, Gregor Thut.

**Validation:** Andra Coldea.

**Visualization:** Andra Coldea.

**Writing – original draft:** Andra Coldea, Monika Harvey, Gregor Thut.

**Writing – review & editing:** Andra Coldea, Stephanie Morand, Domenica Veniero, Monika Harvey, Gregor Thut.

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
