## [Decision Letter · Decision Letter 0]

8 May 2021

PONE-D-21-12956

Parietal alpha tACS shows inconsistent effects on visuospatial attention

PLOS ONE

Dear Dr. Coldea,

Thank you for submitting your manuscript to PLOS ONE. Your paper was reviewed by two experts on the brain stimulation field, both of them found that it is an interesting work and would be an important contribution to the field, however, after careful consideration, we feel that it has merit but does not fully meet PLOS ONE’s publication criteria as it currently stands. The paper suffers from many shortcomings, mainly related to the statistical analysis and to the interpretation of results. Nevertheless,  we invite you to submit a revised version of the manuscript that addresses all points raised during the review process.

We look forward to receiving your revised manuscript.

Kind regards,

Andrea Antal, PhD

Academic Editor

PLOS ONE

Journal Requirements:

Reviewers' comments:

Reviewer's Responses to Questions

**Comments to the Author**

1. Is the manuscript technically sound, and do the data support the conclusions?

Reviewer #1: Yes

Reviewer #2: Yes

2. Has the statistical analysis been performed appropriately and rigorously? 

Reviewer #1: No

Reviewer #2: No

3. Have the authors made all data underlying the findings in their manuscript fully available?

Reviewer #1: No

Reviewer #2: Yes

4. Is the manuscript presented in an intelligible fashion and written in standard English?

Reviewer #1: Yes

Reviewer #2: Yes

5. Review Comments to the Author

Reviewer #1: Review: Parietal alpha tACS shows inconsistent effects on visuospatial attention

Coldea and colleagues show the inconsistency of parietal alpha tACS to modulate visuospatial attention. They replicated a previous paradigm from Schuhmann and colleagues (2019), where after a cue, participants had to indicate whether a gabor patch was oriented clockwise or counterclockwise.

The authors were not able to replicate previous findings, however they did not mirror completely Schuhmann and colleagues (2019) stimulation parameters, in fact they stimulated for 20 instead of 40 minutes and using higher intensity.

General Comment

I like this article. It is easy to follow and I enjoyed reading it. I truly think that null results must be published. However, I have some comments (that need to be addressed in my opinion) and that may help the authors to improve the current version of the manuscript. I did also some suggestion that the authors may ignore if they find them not interesting.

Introduction

Comment

Lines 58-65. Sorry, I do not understand this sentence. You say occipito-parietal tACS at alpha frequency bias perception in a spatially specific manner. Why is it an indirect support that tACS causally interact with attention-related brain oscillation?

Please clarify or expand. (as it is, I do not agree with this sentence, perception and attention are two distinct processes, you can modulate one, without modulating the other process. you do not have to agree with me, just clarify and expand this sentence or change it).

Moreover, I would strongly suggest to expand the sentences “in addition many EEG/MEG- studies have established… alpha power and perception

(most of the paper you cited are about awareness, confidence [instead of perceptual accuracy], response criterion etc.. are you sure that you can say that there is a link between alpha power and perception in general?)

I would be more specific, I think that is established the link between posterior alpha power and TEMPORAL INTEGRATION of visual stimuli. There are a lot of paper about this topic, not only EEG/MEG studies, but also tACS studies and even studies that use sensory entrainment paradigm.

EEG/MEG studies

- Samaha, J., & Postle, B. R. (2015). The speed of alpha-band oscillations predicts the temporal resolution of visual perception. Current Biology, 25(22), 2985-2990.

- VanRullen, R. (2016). Perceptual cycles. Trends in cognitive sciences, 20(10), 723-735.

- Ronconi, L., Oosterhof, N. N., Bonmassar, C., & Melcher, D. (2017). Multiple oscillatory rhythms determine the temporal organization of perception. Proceedings of the National Academy of Sciences, 114(51), 13435-13440.

tACS (Ghiani et al., ( 2021) is an interesting review about tACS and temporal or spatial integration of visual event)

- Battaglini, L., Mena, F., Ghiani, A., Casco, C., Melcher, D., & Ronconi, L. (2020). The effect of alpha tACS on the temporal resolution of visual perception. Frontiers in psychology, 11, 1765.

- Ghiani, A., Maniglia, M., Battaglini, L., Melcher, D., & Ronconi, L. (2021). Binding mechanisms in visual perception and their link with neural oscillations: a review of evidence from tACS. Frontiers in Psychology, 12, 779.

- Cecere, R., Rees, G., and Romei, V. (2015). Individual differences in alpha frequency drive crossmodal illusory perception. Curr. Biol. 25, 231–235.

tACS with focus on the phase

- Ronconi, L., Melcher, D., Junghöfer, M.,Wolters, C. H., and Busch, N. A. (2020). Testing the effect of tACS over parietal cortex in modulating endogenous alpha rhythm and temporal integration windows in visual perception. Europ. J. Neurosci. doi: 10.1111/ejn.15017

Sensory entrainment

- Ronconi, L., & Melcher, D. (2017). The role of oscillatory phase in determining the temporal organization of perception: evidence from sensory entrainment. Journal of Neuroscience, 37(44), 10636-10644.

- Ronconi, L., Busch, N. A., & Melcher, D. (2018). Alpha-band sensory entrainment alters the duration of temporal windows in visual perception. Scientific reports, 8(1), 1-10.

Materials and method

Suggestion:

Please report current density if possible, (it is more important than mA, you can have the same current density with different electrodes size and different intensities)

It would be interesting to see a pictures about voltage distribution if possible.

Results

Why don’t you run an ANCOVA (IAF would be the covariate) when you want to see whether the participants’ performance depends on participants’ IAF instead of grouping participants in an arbitrary way (according to how

much their IAF differed from 10 Hz)? See the analysis conducted by Battaglini et al,. (2020), frontiers in Psychology. Think about it.

- Battaglini, L., Mena, F., Ghiani, A., Casco, C., Melcher, D., & Ronconi, L. (2020). The effect of alpha tACS on the temporal resolution of visual perception. Frontiers in psychology, 11, 1765.

Discussion

Suggestion:

My feeling is that this paper tells us only a part of the tACS story. You focused only in papers that report visuospatial tACS effect and that is ok since it is the main topic. However, in the discussion I would report that parietal cortex was often associated with segmentation process and local processing and that eventually tACS may modulate attentional process only when the task is linked with local processing. (in other words the effect of tACS on perception or attention is task specific)

Please see:

- Zaretskaya, N., & Bartels, A. (2015). Gestalt perception is associated with reduced parietal beta oscillations. Neuroimage, 112, 61-69.

- Battaglini, L., Ghiani, A., Casco, C., & Ronconi, L. (2020). Parietal tACS at beta frequency improves vision in a crowding regime. Neuroimage, 208, 116451.

- Romei, V., Driver, J., Schyns, P. G., & Thut, G. (2011). Rhythmic TMS over parietal cortex links distinct brain frequencies to global versus local visual processing. Current biology, 21(4), 334-337.

Moreover, there is also an interesting study that indicate that parietal cortex is more sensitive to beta activity rather than alpha

- Samaha, J., Gosseries, O., and Postle, B. R. (2017). Distinct oscillatory frequencies underlie excitability of human occipital and parietal cortex. J. Neurosci. 37, 2824–2833. doi: 10.1523/JNEUROSCI.3413-16.2017

Comment

I found interesting the difference in stimulation intensity and duration between yours and Schuhmann and colleagues’ study.

Please consider the phenomenon of stochastic resonance, it works only when the appropriate level of noise is introduced in the system. When you insert too much noise the phenomenon disappears. Analogously, could be that you have inserted in the system too much noise stimulating at 1.5mA?

If you calculate the current density (that is more important than mA) you can find out whether the current that you “inserted in the brain” is equal or greater than previous papers

(current density = mA/cm^2)

Comment

It would be interesting to see how long stimulated previous papers, perhaps tACS need time to produce effect

To the best of my knowledge the papers below (topic: vision and attention) got an effect stimulating for about 40 min.

- Kasten, F. H., Wendeln, T., Stecher, H. I., & Herrmann, C. S. (2020). Hemisphere-specific, differential effects of lateralized, occipital–parietal α-versus γ-tACS on endogenous but not exogenous visual-spatial attention. Scientific reports, 10(1), 1-11.

- Battaglini, L., Mena, F., Ghiani, A., Casco, C., Melcher, D., & Ronconi, L. (2020). The effect of alpha tACS on the temporal resolution of visual perception. Frontiers in psychology, 11, 1765.

- Battaglini, L., Mena, F., Ghiani, A., Casco, C., Melcher, D., & Ronconi, L. (2020). The effect of alpha tACS on the temporal resolution of visual perception. Frontiers in psychology, 11, 1765.

- Schuhmann, T., Kemmerer, S. K., Duecker, F., De Graaf, T. A., Ten Oever, S., De Weerd, P., & Sack, A. T. (2019). Left parietal tACS at alpha frequency induces a shift of visuospatial attention. PLoS One, 14(11), e0217729.

Reviewer #2: The present study of Coldea et al., entitled “Parietal alpha tACS shows inconsistent effects on visuospatial attention”, investigated the behavioural effects of alpha-tACS on spatial attention. Their primary goal is to replicate the reported modulation of spatial attention with alpha-tACS applied to the parietal cortex. They recruited 40 healthy participants who underwent left parietal tACS during task performance in two separate sessions (10 Hz tACS or sham). However, their results indicated that left parietal tACS did not shift the bias to the left, as compared to sham, irrespective of cueing condition. It was an interesting article that will further our understanding of how tACS affects neuronal networks. The results themselves deserve publication because the consistency of brain stimulation after-effects is a matter of recent debate. However, there are some points that I want clarifications and suggestions that the authors may consider to improve the quality of the manuscript.

Methods

1. On page 10, kindly (shortly) described how trials are considered “valid”, “invalid”, and “neutral”.

2. Indicate the stimulation duration in the “tACS section”.

3. “Participants were instructed to respond as fast and as accurately as possible, using the index and middle finger of their right hand”…..did they use button presses? How is the reaction time (RT) defined?

Data analysis

1. Is the statistical analysis of the data entirely identical to Schuhmann et al., 2019? RTs are not normally distributed, did the authors considered using non-parametric tests or log-transformed them before performing their parametric tests (ANOVA)?

Results

1. The authors reported a significant main effect of cue validity (Figure 2A). However, by looking in Figure 2A, we can clearly see that the differences between the valid trials (450.7±70.5ms), neutral trials (464±76ms) and invalid trials (485.7±85.8ms) are not robust. The overlapping SDs are also indicative. Are the post hoc comparisons corrected? For example Bonferroni? Kindly check your data for outliers.

2. I have the same opinion for Fig 2B. I would not assume significant differences between the hemifields. Why ANOVA? You are only comparing two factors, right and left hemifield.

3. The analysis for the State-dependency of tACS-effects (on RT): Pre-existing spatial bias seems erroneous. The authors considered the change in spatial bias from sham to tDCS. But this scenario only validly applied for those participants that were stimulated with sham on the first session and tACS on the second session. I meant the “order effect” because the session was counterbalanced (sham-tACS vs tACS-sham). In my opinion, the two-day intersession interval is short, so we cannot rule out the cross-over effect. Therefore we cannot correctly consider the spatial bias from subjects with a tACS-sham session schedule as pre-existing. This comment also applies to the rest of the analysis in this section (IAF, amplitude).

4. Of the 40 participants whose data were included in the analysis, 25 could distinguish between 10 Hz tACS and sham sessions correctly.

This indicates that blinding was compromised in more than half of the participants. Is this comparable to the previous studies? I want advice analysing the date of the group which remained blinded.

6. PLOS authors have the option to publish the peer review history of their article (what does this mean?). If published, this will include your full peer review and any attached files.

Reviewer #1: **Yes: **Luca Battaglini

Reviewer #2: No

---

## [Author Response · Author response to Decision Letter 0]

5 Jul 2021

We are grateful to the editors and reviewers for taking the time to read and provide constructive suggestions for improvement of our manuscript. We have revised our manuscript accordingly, taking into consideration all comments. We have addressed all the comments point-by-point and explained how we have integrated them in the Response to Reviewers letter, and have highlighted the changes we made to the original manuscript.

Yours Sincerely,

On behalf of the co-authors

Andra Coldea

---

## [Decision Letter · Decision Letter 1]

16 Jul 2021

Parietal alpha tACS shows inconsistent effects on visuospatial attention

PONE-D-21-12956R1

Dear Dr. Coldea,

We’re pleased to inform you that your manuscript has been judged scientifically suitable for publication and will be formally accepted for publication once it meets all outstanding technical requirements.

Kind regards,

Andrea Antal, PhD

Academic Editor

PLOS ONE

Additional Editor Comments (optional):

Reviewers' comments:

Reviewer's Responses to Questions

**Comments to the Author**

1. If the authors have adequately addressed your comments raised in a previous round of review and you feel that this manuscript is now acceptable for publication, you may indicate that here to bypass the “Comments to the Author” section, enter your conflict of interest statement in the “Confidential to Editor” section, and submit your "Accept" recommendation.

Reviewer #1: All comments have been addressed

Reviewer #2: All comments have been addressed

2. Is the manuscript technically sound, and do the data support the conclusions?

Reviewer #1: Yes

Reviewer #2: Yes

3. Has the statistical analysis been performed appropriately and rigorously? 

Reviewer #1: Yes

Reviewer #2: Yes

4. Have the authors made all data underlying the findings in their manuscript fully available?

Reviewer #1: Yes

Reviewer #2: No

5. Is the manuscript presented in an intelligible fashion and written in standard English?

Reviewer #1: Yes

Reviewer #2: Yes

6. Review Comments to the Author

Reviewer #1: The authors have sufficently addressed all of my comments. Thank you for this careful revision.

(one last point, please see at line 535, you say "All relevant pre-processed data are available on Reshare at:" but I cannot see the link)

Reviewer #2: (No Response)

7. PLOS authors have the option to publish the peer review history of their article (what does this mean?). If published, this will include your full peer review and any attached files.

Reviewer #1: No

Reviewer #2: **Yes: **Shane Fresnoza MD PhD

---

## [Editor Report · Acceptance letter]

27 Jul 2021

PONE-D-21-12956R1 

Parietal alpha tACS shows inconsistent effects on visuospatial attention 

Dear Dr. Coldea:

I'm pleased to inform you that your manuscript has been deemed suitable for publication in PLOS ONE. Congratulations! Your manuscript is now with our production department. 

Kind regards, 

on behalf of

Prof. Dr. Andrea Antal 

Academic Editor

PLOS ONE